# The Role of Circulating MicroRNAs in Patients with Early-Stage Pancreatic Adenocarcinoma

**DOI:** 10.3390/biomedicines9101468

**Published:** 2021-10-14

**Authors:** Michal Eid, Paraskevi Karousi, Lumír Kunovský, Štěpán Tuček, Dagmar Brančíková, Zdeněk Kala, Ondřej Slabý, Jiří Mayer, Christos K. Kontos, Jan Trna

**Affiliations:** 1Department of Hematology, Oncology and Internal Medicine, University Hospital Brno, Faculty of Medicine, Masaryk University, 62500 Brno, Czech Republic; eid.michal@fnbrno.cz (M.E.); tucek.stepan@fnbrno.cz (Š.T.); brancikova.dagmar@fnbrno.cz (D.B.); mayer.jiri@fnbrno.cz (J.M.); 2Department of Biochemistry and Molecular Biology, Faculty of Biology, National and Kapodistrian University of Athens, 15701 Athens, Greece; pkarousi@biol.uoa.gr; 3Department of Gastroenterology and Internal Medicine, University Hospital Brno, Faculty of Medicine, Masaryk University, 62500 Brno, Czech Republic; kunovsky.lumir@fnbrno.cz; 4Department of Surgery, University Hospital Brno, Faculty of Medicine, Masaryk University, 62500 Brno, Czech Republic; kala.zdenek@fnbrno.cz; 5Department of Biology, Faculty of Medicine, Masaryk University, 62500 Brno, Czech Republic; oslaby@med.muni.cz; 6Central European Institute of Technology, Masaryk University, 62500 Brno, Czech Republic; 7Department of Gastroenterology and Digestive Endoscopy, Masaryk Memorial Cancer Institute Brno, 65653 Brno, Czech Republic

**Keywords:** pancreatic cancer, early stage, microRNA, diagnosis, prognosis, chemoresistance

## Abstract

Pancreatic ductal adenocarcinoma (PDAC) is increasing in incidence and is still associated with a high rate of mortality. Only a minority of patients are diagnosed in the early stage. Radical surgery is the only potential curative procedure. However, radicality is reached in 20% of patients operated on. Despite the multidisciplinary approach in resectable tumors, early tumor recurrences are common. Options on how to select optimal candidates for resection remain limited. Nevertheless, accumulating evidence shows an important role of circulating non-coding plasma and serum microRNAs (miRNAs), which physiologically regulate the function of a target protein. miRNAs also play a crucial role in carcinogenesis. In PDAC patients, the expression levels of certain miRNAs vary and may modulate the function of oncogenes or tumor suppressor genes. As they can be detected in a patient’s blood, they have the potential to become promising non-invasive diagnostic and prognostic biomarkers. Moreover, they may also serve as markers of chemoresistance. Thus, miRNAs could be useful for early and accurate diagnosis, prognostic stratification, and individual treatment planning. In this review, we summarize the latest findings on miRNAs in PDAC patients, focusing on their potential use in the early stage of the disease.

## 1. Introduction

Pancreatic cancer is the 11th most common cancer, while nearly 90% of all pancreatic tumors are characterized as pancreatic ductal adenocarcinoma (PDAC) [1]. Based on the GLOBOCAN 2018 data, almost 460,000 new cases of PDAC were diagnosed worldwide, while the mortality rate is still high and almost identical to the incidence [2]. No milestone treatment has been implemented into therapeutic options, in recent years. Thus, radical resection is the only curative approach; however, whether radical resection is possible or not, is mainly determined by perioperative findings.

Due to the absence of specific symptoms, the majority of patients with PDAC are diagnosed in the advanced stage, resulting in poor prognosis. Additionally, resection with curative intent is indicated only in a few of the cases, while surgery is associated with a high postoperative morbidity rate [3]. Disease relapses are also frequent, as PDAC has a high tendency of dissemination even at the early stage [4].

It is, therefore, obvious that one of the main goals of current research is the early diagnosis PDAC, or even detection of precancerous pancreatic lesions, as the risk of micrometastatic dissemination is the lowest one. For PDAC patients, there is neither recommended population-based screening program nor highly sensitive and specific diagnostic biomarkers. The most widely used biomarker in clinical practice is carbohydrate antigen 19-9 (CA 19-9). However, as PDAC is usually asymptomatic at the early stage, the positive predictive value of CA 19-9 is only 0.9% in this setting [5]. Currently, attempts have been made towards the incorporation of artificial intelligence (AI) tools to further support the PDAC early detection efforts; however, this field is still in its infancy and requires multidisciplinary approaches to evolve [6].

Next to the accurate approach enabling diagnosis of PDAC at an early stage, prediction of patients’ survival is another important issue. Currently, multiple prognostic factors are available in daily practice; radiographical staging, performance status, tumor grade, perineural and lymphovascular infiltration, resection margin status, CA 19-9 level, or inflammatory markers are the most common prognosticators [7,8]. However, their prognostic accuracy is limited.

Potential molecular biomarkers for early diagnosis and prognostic stratification include circulating microRNAs (miRNAs). Nowadays, there is no doubt that miRNAs play a crucial role in both pathological and physiological processes via post-transcriptional regulation of gene expression [9,10,11]. A single miRNA may target hundreds of messenger RNAs (mRNAs) [12]. Down- or up-regulation of specific miRNAs may dysregulate the activity of a target oncogene or tumor suppressor gene. Thus, according to the final effect, miRNAs are divided into the group of oncogenic miRNAs (oncomiRs) and the group of tumor-suppressor miRNAs [13,14,15]. Emerging evidence suggests that miRNAs are heavily involved in the process of carcinogenesis, including proliferation, survival, invasion, and metastasis [16,17]. Next to the diagnostic and prognostic significance, different expression levels of specific miRNAs may also be used for the prediction of chemoresistance and facilitate personalized treatment planning [18].

The purpose of this systematic review is to summarize the current knowledge about the potential use of circulating miRNAs as molecular biomarkers for diagnosis, prognosis, and chemoresistance prediction concerning early-stage PDAC.

## 2. Biogenesis of Circulating miRNAs

miRNAs are endogenous small RNAs with a length of 18–25 nucleotides [19]. The transcription of miRNA genes is independent and regulated by their own promoters. Several miRNAs genes are located within an intron or untranslated region (UTR) of a protein-coding gene. These miRNAs are transcribed along with their host genes [14,20,21]. Most miRNAs are transcribed in the nucleus into primary miRNAs (pri-miRNAs) by RNA polymerase II [22,23,24]. Subsequently, the processing of the pri-miRNA by the DROSHA-DGCR8 complex leads to the formation of 70-nucleotide long precursor miRNAs (pre-miRNAs). Then, Exportin 5 transfers the pre-miRNAs from the nucleus to the cytosol. The subsequent development of mature miRNAs is achieved after the cleavage of the pre-miRNA by the ribonuclease DICER; this cleavage leads to the formation of a miRNA duplex with an approximate length of 22 nucleotides (Figure 1) [25,26,27,28]. One of the duplex strands is loaded onto the RISC complex and acts as the mature miRNA. Their mechanism of regulation of gene expression is based on the interaction of their seed sequence with a specific sequence, mostly at the 3′ UTR of target mRNAs, but target sequences are found at the 5′ UTR or even within the coding regions of mRNAs, as well [29,30,31]. This interaction induces translational repression or mRNA deadenylation and degradation, provided that perfect complementarity between the miRNA and the mRNA is achieved [19,32].

For the isolation and detection of selected miRNAs, cancer tissue is usually used. However, in PDAC patients, diagnosis may be limited on the cytology findings from the fine-needle aspiration biopsy. This limitation does not apply to blood samples, in which circulating miRNAs are determined. In the pancreatic tumor mass, cancer cells contribute to the pool of circulating cancer-specific miRNAs in blood [33]. miRNAs can be secreted to the circulation from cells within exosomes, which are extracellular vesicles that contain proteins, DNA, and RNA. These exosomes protect circulating miRNAs from degradation by RNases and are taken up by other cells, the function and behavior of which may be affected by the content carried by the exosomes. Thus, exosomes are involved in the intercellular communication and development of cancer [34]. Circulating miRNAs can also be exosome-free and are associated with Ago2 forming ribonucleoprotein complexes, which protect them from degradation by nucleases in biological fluids [35,36,37]. Moreover, Vickers et al. revealed another mechanism of miRNA intercellular transport in human plasma. Some miRNAs were found in purified fractions of high-density lipoprotein (HDL), which is able to deliver endogenous miRNAs to recipient cells. However, this transport mechanism transfers only a minor proportion of the total circulating miRNAs (Figure 1) [36,38]. Besides tumor tissue, blood plasma, and serum, miRNAs can also be found in saliva, urine, and other body fluids [39].

## 3. Methods for Detection of Circulating miRNAs

Current research mainly depends on samples that are directly acquired from pancreatic tissue. Naturally, surgical resection provides enough material for analysis. Unfortunately, resection is feasible only in the minority of pancreatic cancer cases. The remaining patients undergo endosonography with fine-needle aspiration biopsy. During this diagnostic procedure, a biopsy can be taken for further assessment. There are several disadvantages to this approach. For molecular testing, the amount of biologic material can be limited; furthermore, contamination of samples by blood or surrounding non-malignant cells is possible [40]. Moreover, the number of false-negative biopsies can be up to 4% and the negative predictive value is 85% [41].

Blood samples are also suitable for the assessment of diagnosis and determination of prognosis. Currently, there are three main methods of quantifying circulating miRNAs: real-time quantitative polymerase chain reaction (real-time qPCR), gene expression arrays, and sequencing. The most frequently used method is qPCR since it is a simple, cost-effective, and reliable method. Nevertheless, several other limitations must be considered when circulating miRNA levels are analyzed. Besides the sufficient amount of miRNAs in serum or plasma samples, attention must also be paid to specimen preservation methods and time, centrifugation steps, miRNA extraction, and normalization methods [42,43,44,45].

Most studies that have aimed so far at quantifying circulating miRNAs in pancreatic cancer patients’ samples use commercially available kits, specific for RNA extraction from plasma; however, TRIzol-based methods, suitable for isolating circulating miRNAs, have also been described [46]. In some cases, synthetic miRNAs of *Caenorhabditis elegans* are added as spike-in controls to the plasma and/or serum samples prior to RNA extraction, in order to evaluate the efficiency of the extraction and use them for normalization of human miRNA levels in the following steps [45,47].

In qPCR assays, after obtaining an RNA extract, a reverse transcription step follows, in order to synthesize first-strand cDNA. In some studies, a polyadenylation step is carried out before reverse transcription, in order to add a poly(A) tail to each miRNA [46]. Next, first-strand cDNA is used as a template to conduct qPCR, using specific primers or probes to detect miRNAs. The controls used for normalization vary among studies, from the aforementioned spike-in controls to endogenous ribosomal RNAs, small nucleolar RNAs, or another miRNA, the levels of which are considered not to vary among samples [46,48,49]. The selection of the reference molecule is a crucial step for proper quantification.

Gene arrays are used to detect the expression of multiple miRNAs at the same time. Array analysis in samples is useful for interpreting the miRNA expression profiles of normal and tumor tissues. When using arrays, the RNA extract is labeled, frequently with biotin, and then injected into an array chip to hybridize with the respective fixed complementary DNA sequences. Subsequently, the arrays are washed and scanned for signal detection, prior to bioinformatic analysis [50,51]

Last, next-generation sequencing has also been implemented to generate circulating miRNA profiles in patients’ blood samples [52]; however, due to its high cost, it is often not preferred to qPCR.

## 4. Circulating miRNAs and Their Diagnostic Significance in Early-Stage PDAC

As already mentioned, at the early stage of PDAC, there are no specific warning clinical symptoms. Thus, the majority of patients are diagnosed at an advanced stage, when the tumor is symptomatic and unresectable. However, multiple studies among PDAC patients have already identified significant miRNA signatures with diagnostic efficiency at early-stage PDAC, or even in precancerous pancreatic lesions [53]. Moreover, a miRNA expression profile may distinguish between malignant and benign lesions in pancreatic tissue [54]. These data render circulating miRNAs very promising as molecular biomarkers.

Duell et al. published a prospective cohort study evaluating the relative expression of a panel of plasma miRNAs. Samples were collected years before the diagnosis of PDAC within the Prospective Investigation into Cancer and Nutrition cohort (EPIC). This study included 521,457 participants from ten European countries. Participants had no previous cancer history and most of them were enrolled between 1992 and 1998 and at the age of 35–70 years. When PDAC was histologically confirmed, alive and cancer-free normal controls from the EPIC cohort were randomly selected and subsequently matched to each PDAC case, based on several factors, including sex, study center/country, age at blood sample collection, and date of sampling. The final cohort consisted of 225 PDAC cases (localized: 13; metastatic: 96; missing data on stage: 116) and 225 matched normal controls. The median follow-up time between blood sample collection and diagnosis of PDAC was 7.85 years. Increased concentration of miR-21-5p, miR-10b-5p, and miR-30c-5p was associated with shorter survival time intervals (≤5 years). For longer follow-up times (≤12 years), the most significant increase was observed in the levels of miR-10a-5p, miR-10b-5p, and miR-30c-5p. The results of this study provide support for the hypothesis that the upregulation of specific miRNAs may indicate individuals who may be at a higher risk of developing PDAC [55].

The biomarker utility of other circulating miRNAs was also evaluated in early-stage PDAC. Xue et al. analyzed 29 studies assessing the potential of circulating miRNAs as non-invasive diagnostic biomarkers. From a total of 68 evaluated miRNAs, 51 were analyzed as individual ones. Ten studies reported 13 miRNA panels, each of which contains 2–15 miRNAs [56]. For miRNA detection and quantification, qPCR was performed in all studies. The normalization methods for the expression of miRNAs were not the same. However, 11 individual miRNAs (miR-10b-5p, miR-20a-5p, miR-21-5p, miR-22-3p, miR-30c-5p, miR-106b-5p, miR-122-5p, miR-181a-5p, miR-642b-3p, miR-885-5p, and let-7a-5p) and 3 panels (panel A: miR-196a-5p, and miR-196b-5p; panel B: miR-22-3p, miR-642b-3p, and miR-885-5p; panel C: miR-20a-5p, miR-21-5p, miR-24-3p, miR-25-3p, miR-99a-5p, miR-185-5p, and miR-191-5p) showed excellent sensitivity and specificity (both ≥90%) for all stages of [56,57,58,59]. Another panel of seven selected miRNAs (miR-20a-5p, miR-21-5p, miR-24-3p, miR-25-3p, miR-99a-5p, miR-185-5p, and miR-191-5p) from PDAC patients’ serum was compared to normal controls. All seven miRNAs were over-represented more than two-fold. In resectable tumors, the positive detection rate was 96% for stage I and 91.7% for stage II. These results demonstrate that this panel could have non-invasive diagnostic biomarker utility in early-stage PDAC [60]. Similarly, Ganepola et al. demonstrated high sensitivity (91%) and specificity (91%) of a diagnostic panel consisting of three plasma miRNAs (miR-642b-3p, miR-885-5p, and miR-22-3p) in 11 patients with early-stage PDAC [61].

A study conducted by Xu et al. identified 13 deregulated plasma miRNAs in PDAC patients compared to normal controls. These miRNAs were further validated in a multicenter trial with a cohort of 363 subjects. Once plasma miRNAs levels in PDAC patients were compared to those of normal controls, significant differences were observed for miR-486-5p, miR-126-3p, and miR-938 levels. Furthermore, the levels of miR-126-3p, miR-26b-3p, miR-938, and miR-19b-3p were significantly different in PDAC vs. pancreatic neuroendocrine tumors. Moreover, the panel of miR-486-5p, miR-126-3p, miR-938, miR-663b, and miR-19b-3p was able to discriminate PDAC patients from patients with chronic pancreatitis (CP). On the other hand, only miR-938 had a significant diagnostic value for PDAC vs. other pancreatic tumors [62].

In a study conducted by Khan et al., 125 serum samples were analyzed by qPCR. In PDAC patients, 3 miRNAs (miR-215-5p, miR-122-5p, and miR-192-5p) were significantly upregulated, compared to CP patients. In addition, miR-30b-5p and miR-320b were significantly downregulated in PDAC patients’ serum, compared to CP patients and normal controls. The results from this study also demonstrated that these five miRNAs had significantly different levels between the serum of patients with CP and those with early-stage PDAC. This panel has the potential to become part of a multicomponent screening panel of CP patients for PDAC [63].

### 4.1. Characteristics of the Most Studied miRNAs with Diagnostic Potential

#### 4.1.1. miR-21-5p

One of the most promising and investigated miRNAs in PDAC is miR-21-5p, which is also overexpressed in many other types of cancer [64]. It is commonly considered as a miRNA with an oncogenic role, targeting several tumor suppressor genes, such as phosphatase and tensin homolog (PTEN), tropomyosin 1 (TM1), programmed cell death 4 (PDCD4), or tissue inhibitor of metalloproteinase 3 (TIMP3) [65,66]. Thus, its high expression is associated with anti-apoptotic activity, proliferation, migration, invasion, and survival of cancer cells [67,68].

In PDAC, its potential use as a diagnostic biomarker is supported by a high number of studies and it can be assumed that miR-21-5p is likely to be a part of the diagnostic miRNA panel [69]. In addition, miR-21-5p is upregulated even in microscopic precancerous pancreatic lesions such as non-invasive pancreatic intraepithelial neoplasia [70,71,72,73], compared to normal controls. Furthermore, in pancreatic benign lesions, such as serous cystadenomas and intraductal papillary mucinous neoplasms (IPMNs), which have the potential for a malignant transformation, upregulation of miR-21-5p has also been recorded [74,75]. Similar results were reported in a study by Abue et al., in which the levels of miR-21-5p in plasma were compared between patients with PDAC, IPMN, and normal controls. The mean miR-21-5p levels were significantly higher in PDAC patients’ plasma as well as in the plasma of patients with IPMN, compared to normal controls [76].

#### 4.1.2. miR-25-3p

Upregulation of miR-25-3p suppresses PH domain leucine-rich repeat protein phosphatase 2 (PHLPP2) with subsequent activation of oncogenic AKT/mTOR/p70S6K (RPS6KB2) signaling pathway inducing carcinogenesis and proliferation of PDAC cells [77]. The diagnostic value of serum miR-25-3p was evaluated in a case-control study by Yu et al. Eighty patients with PDAC and 91 normal controls were enrolled. In PDAC cases, miR-25-3p levels were significantly higher than in normal controls. When combined with CA 19-9, the diagnostic performance was high, showing a sensitivity of 97.5%, and a specificity of 90.1%. At the early stage, the sensitivity of this combination of biomarkers was significantly higher than CA 19-9 alone, which, as already mentioned, is commonly used in the diagnostic process. No association between miR-25-3p levels and disease stage was found. Further examination is required to validate these findings [78].

#### 4.1.3. miR-182-5p

One of miR-182-5p targets is beta-transducing repeat-containing protein (β-TrCP2; also known as FBXW11), which plays an important role in the ubiquitin–proteasome system regulating cellular homeostasis. In addition, β-TrCP2 targets phosphorylated β-catenin and thus regulates the Wnt/β-catenin pathway [79,80]. Aberrant Wnt/β-catenin signaling is implicated in pancreatic carcinogenesis and is common in PDAC [81]. Upregulation of miR-182-5p is associated with accelerated proliferation, invasion, and migration of tumor cells [82].

In a study by Chen et al., plasma miR-182-5p levels were significantly higher in PDAC patients than in CP patients and normal controls. Moreover, in the group of PDAC patients, miR-182-5p levels were also associated with clinical stages. In stage IV, its levels were higher than those at stages I, II, and III. Thus, miR-182-5p levels could indicate the disease stage. When plasma miR-182-5p was combined with the widely used biomarker CA 19-9 in a diagnostic setting, sensitivity and specificity were 84.7% and 86.8%, respectively. Results of this study indicate that circulating miR-182-5p may serve as a novel, surrogate diagnostic biomarker for PDAC [83].

#### 4.1.4. miR-221-3p

The oncogenic role of miR-221-3p is indicated by the fact that its higher levels in PDAC have an anti-apoptotic activity and promote cell proliferation and invasion by inducing the expression of matrix metallopeptidases 2 (MMP2) and 9 (MMP9). Direct targets of miR-221-3p are PTEN and TIMP2 [84,85]. In the study of Kawaguchi et al., plasma miR-221-3p levels were significantly higher in PDAC patients than in normal controls [48].

#### 4.1.5. miR-483-3p

miR-483-3p significantly represses the expression of the tumor-suppressor *SMAD4*. Its mutation or deletion is detected in more than 50% of PDAC cases and is associated with aggressive carcinogenesis [86,87]. In a study by Abue et al., the mean levels of miR-483-3p in plasma were significantly higher in PDAC patients than in patients with IPMN or normal controls [76]. These results indicate that miR-483-3p is a promising diagnostic biomarker in PDAC.

#### 4.1.6. miR-10b-5p

In PDAC, miR-10b-5p is commonly upregulated and its levels in plasma are significantly increased, in comparison with normal controls. Its value is both diagnostic and prognostic. Tat-interacting protein 30 (TIP30) downregulation by miR-10b-5p in pancreatic cancer cells was shown to lead to enhanced EGFR activity and downstream activation of extracellular signal-regulated kinases 1 and 2 (ERK1/2) [88,89]. Some studies have shown that EGFR is highly expressed in pancreatic cancer. High EGFR expression has been associated with advanced disease, poor survival, and the presence of metastases [90].

Expression levels of miR-10b-5p in plasma were quantified in 17 PDAC patients, 5 CP patients, and 20 normal controls. In PDAC patients, these levels were increased 575-fold when compared to the corresponding levels in either CP patients or normal controls [89]. Plasma levels of miR-10b-5p had 100% sensitivity and specificity in PDAC, distinguishing it from normal controls [58]. A summary of circulating miRNAs with a diagnostic value is presented in Table 1.

## 5. Circulating miRNAs and Their Prognostic Significance in Early-Stage PDAC

Currently, the prediction of survival of patients with early-stage PDAC is limited on several factors such as disease stage, resection margin status, postoperative levels of CA 19-9, and performance status. The R0 resection status is an important prognostic factor indicating the probability of long survival and is the goal of a multimodal approach with regard to resectable tumors [91]. Considering the toxicity of neoadjuvant chemotherapy/chemoradiotherapy, morbidity, and mortality of surgical treatment, patients should be very carefully selected.

The expression of specific miRNAs has the potential to become a useful biomarker for the accurate prediction of prognosis and the choice of an optimal therapeutic approach. Similar to other types of cancer, a specific miRNA signature has been shown to hold an important prognostic value, being able to define cancer features such as malignant potential, histological grade, tumor stage and aggressiveness, and cancer cell proliferation index [92].

### 5.1. Characteristics of the Most Studied miRNAs with Prognostic Value

It is difficult to evaluate the prognostic role of individual miRNAs, as they are analyzed in studies particularly within panels, in which they will most likely be used in the future. Thus, the following paragraphs summarize the current knowledge about miRNAs that represent promising prognostic biomarkers, either as standalone or combined. A brief summary of deregulated miRNAs with a prognostic significance is presented in Table 2.

#### 5.1.1. miR-21-5p

Besides being considered as a diagnostic marker, high plasma miR-21-5p levels are considered as a negative prognostic marker, significantly associated with advanced disease stages, lymph node infiltration, liver metastases, and worse survival, compared to a patient group with low miR-21-5p levels [76]. Similar conclusions were drawn by the metanalysis of Hu et al. Expression of miR-21-5p was measured by either qPCR [60,65,93,94,95,96,97,98,99] or in situ hybridization [100,101,102]. Samples used for analysis included fresh frozen tissues, formalin-fixed paraffin-embedded tissues, and blood sera. Results indicate that elevated levels of miR-21-5p are significantly associated with worse survival. In subgroup analyses, the same conclusions were drawn from serum-based studies [60,99,102,103].

#### 5.1.2. miR-375-3p

miR-375-3p is usually associated with tumor suppression and downregulated in many malignancies. miR-375-3p targets oncogenes, such as astrocyte elevated gene 1 (AEG1), pyruvate dehydrogenase kinase 1 (PDK1), Janus kinase 2 (JAK2), and insulin-like growth factor 1 receptor (IGF1R). However, several studies demonstrated its overexpression in certain tumors, such as prostate cancer [106].

The results from a study conducted by Karasek et al. among 25 patients with resectable PDAC highlighted the prognostic value of a panel of miR-375-3p and miR-21-5p. Their higher preoperative levels were significantly associated with worse overall survival (OS). Moreover, miR-21-5p concentration in plasma was shown to be independent of other clinicopathological factors [104].

#### 5.1.3. miR-365a-3p and miR-99a-5p

Upregulated miR-365a-3p inhibits the function of the nuclear factor-κB (NF-κB) by downregulating NF-κB subunit REL, thus inducing apoptosis and leading to a decrease in the viability of PDAC cells. miR-99a-5p has a role in many human malignancies and its aberrant expression has been linked with either oncogenic or tumor-suppressive activity. miR-99a-5p affects pancreatic cancer cell migration and invasion by regulating the mammalian target of rapamycin (mTOR) [107].

Gablo et al. demonstrated that high preoperative levels of miR-365a-3p and miR-99a-5p predict longer survival after curative resection in PDAC patients. Levels of both molecules in patients with longer (OS > 20 months) and shorter (OS < 16 months) survival after resection were significantly different. These miRNAs may distinguish early-stage PDAC patients having no benefit from the surgery, with regard to survival. In the same study, miR-200c-3p was identified as another potential prognostic marker, as its significantly higher levels are associated with poor prognosis [19].

#### 5.1.4. miR-182-5p

The prognostic value of miR-182-5p in PDAC has been evaluated by Chen et al. In total, 109 patients with PDAC were enrolled in this study, and 38 of them were at stage I+II. Patients with resected PDAC were divided into two groups, those with low circulating miR-182-5p levels and those with high levels, based on the determined cut-off value. Higher miR-182-5p levels in plasma were significantly associated with shorter DFS and OS [83].

#### 5.1.5. miR-196a-5p

miR-196a-5p downregulates inhibitor of growth 5 (ING5) and is upregulated in pancreatic cancer. It is associated with impaired apoptosis, increased proliferation, and invasiveness of cancer cells [60].

miR-196a-5p showed prognostic significance in the study of Kong et al. High serum miR-196a-5p levels were significantly associated with inferior median OS. Moreover, serum miR-196a-5p levels were higher in patients with PDAC at unresectable stages (III and IV) than in those with early-stage PDAC. This finding could be exploited in a personalized therapy approach [105]. Similarly, Bloomston et al. observed that high tissue expression levels of miR-196a-5p among resected PDAC samples were significantly associated with inferior median OS, compared to low levels (14.3 vs. 26.5 months) [88].

#### 5.1.6. miR-221-3p

The prognostic value of miR-221-3p was also highlighted. In PDAC patients, high plasma miR-221-3p levels were significantly associated with the presence of distant metastasis and unresectable status. Furthermore, plasma miR-221-3p levels in postoperative samples were significantly reduced, compared to matched preoperative samples. Thus, miR-221-3p may be useful as a marker of recurrence of PDAC after tumor resection [48].

## 6. miRNAs as a Marker of Chemoresistance

Several clinical trials have demonstrated a significant survival benefit of adjuvant treatment of patients with early-stage PDAC. Despite the intensity of systemic chemotherapy causing a decline in quality of life due to toxicity effects, little is known about the features of the resectable tumors that are related to patients’ survival. PDAC generally shows low responsiveness to chemotherapy, resulting in a poor prognosis [108]. The mechanisms responsible for the chemoresistance are not well known, but there are two main types of drug resistance—intrinsic, when the tumor is not sensitive to chemotherapy before the start of treatment; and acquired, when the tumor acquires resistance during the therapy but initially was sensitive. The intrinsic type of resistance may be caused by physiological barriers limiting the absorption of the drug into certain tissues, such as dense desmoplastic stroma and hypovascularity, which are typical for PDAC [109]. The acquired resistance is commonly associated with the expression of energy-dependent transporters that are able to detect and clear cytostatics from cancer cells. These drug efflux pumps keep the intracellular level of cytostatics low. This mechanism is associated with multidrug resistance. In addition, there are other mechanisms (such as increased DNA repair) that are related to genetic and epigenetic changes in cancer cells [110,111].

It has been demonstrated that miRNAs may also be involved in the process of chemoresistance due to their ability to modulate drug efflux, cell cycle, and apoptotic response. miRNAs regulate ATP-binding cassette (ABC) membrane transporters, responsible for transporting drugs outside of the cell [25,112]. However, the majority of studies demonstrating the effect of miRNAs on chemoresistance are based on intracellular miRNAs of PDAC cells. Concerning circulating miRNAs, the effect on chemoresistance needs to be confirmed in further studies that may contribute to the optimization of treatment approaches, especially when neoadjuvant chemotherapy is considered.

Gisel et al. observed dysregulation of miR-181a-5p and miR-218-5p in chemoresistant PDAC cell lines with P-glycoprotein overexpression [113]. Decreased plasma miR-181a-5p levels were associated with a good response to FOLFIRINOX in advanced PDAC [114]. Cancer cell proliferation can be decreased by overexpression of miR-192-5p and miR-215-5p. This may have a negative impact on the effectiveness of drugs specifically targeting the S-phase of the cell cycle [115].

Several authors published results demonstrating that elevated expression of miR-21-5p may decrease the antitumor effect of 5-fluorouracil and gemcitabine [116,117]. miR-21-5p inhibits the tumor suppressor PTEN with subsequent activation of PI3K/AKT/mTOR signaling pathway, thus decreasing susceptibility of cancer cells to apoptosis [94]. Conversely, sensitivity to gemcitabine may be increased by the inhibition of miR-21-5p indicating that miRNAs have the potential to become a promising target within a personalized therapeutic approach [118].

miR-7-5p acts as a tumor suppressor and inhibits multiple oncogenic targets in many cancer types. Inhibition of PDAC progression may be caused by impairment of autophagy-derived pools of glucose. Ye et al. quantified miR-7-5p levels in the serum of patients with stage III or IV PDAC. Unsurprisingly, the expression levels were decreased in PDAC patients, compared to normal controls. Moreover, sensitivity to gemcitabine was strongly associated with miR-7-5p levels. In opposite to this, in vitro upregulation of miR-7-5p significantly improved the sensitivity of PDAC cells to gemcitabine [15].

These results indicate that miRNAs could serve as molecular biomarkers assisting decision-making for neoadjuvant chemotherapy or primary tumor resection in patients with early-stage PDAC. Targets of specific miRNAs and their biological function are illustrated in Figure 2.

## 7. Conclusions

Only a modest improvement in PDAC survival was observed during the last 30 years. Due to the asymptomatic early stage and late diagnosis of PDAC, the lack of diagnostic biomarkers, and the general aggressiveness of PDAC, only a small proportion of patients are candidates for curative resection. The increasing knowledge about the molecular mechanisms of carcinogenesis indicates the importance of individualized treatment planning in many types of cancer [119]. Circulating miRNAs represent promising diagnostic and prognostic biomarkers in PDAC. Their main advantages include impressive stability in body fluids as well as tissue- and disease-specific expression. Moreover, multiple studies demonstrated their potential in the prediction of chemosensitivity. Panels of certain appropriately selected circulating miRNAs may improve the sensitivity and specificity and may become an attractive non-invasive tool.

In addition, specific inhibition of oncogenic miRNAs in mice by intravenously administrated antagomiRs resulted in a reduction of the expression levels of corresponding miRNAs in many organs [120]. Concomitant antisense inhibition of miR-21-5p and miR-221-3p in tumor-initiating stem-like cells in pancreatic cancer modulates carcinogenesis, arrests the cell cycle, induces apoptosis, decreases the metastatic potential, and increases chemosensitivity [118,121]. This potential new therapeutic approach may significantly contribute to a strategy of personalized medicine in PDAC patients.

However, clinical studies conducted to date are not uniform in terms of the patient population or clinical stage of the disease. Moreover, no consensus exists regarding the standardization of the pre-analytical, analytical, and post-analytical stages. Thus, there are discrepancies in the method of collection, the time span between sample collection and centrifugation, RNA isolation protocols, purity assessment, choice of miRNA detection platform, and normalization method used for quantification. This is considered to be one of the major limitations for the successful implementation of circulating miRNAs in clinical practice. Large prospective trials and case-control studies are needed for this purpose.

## Figures and Tables

**Figure 1 biomedicines-09-01468-f001:**
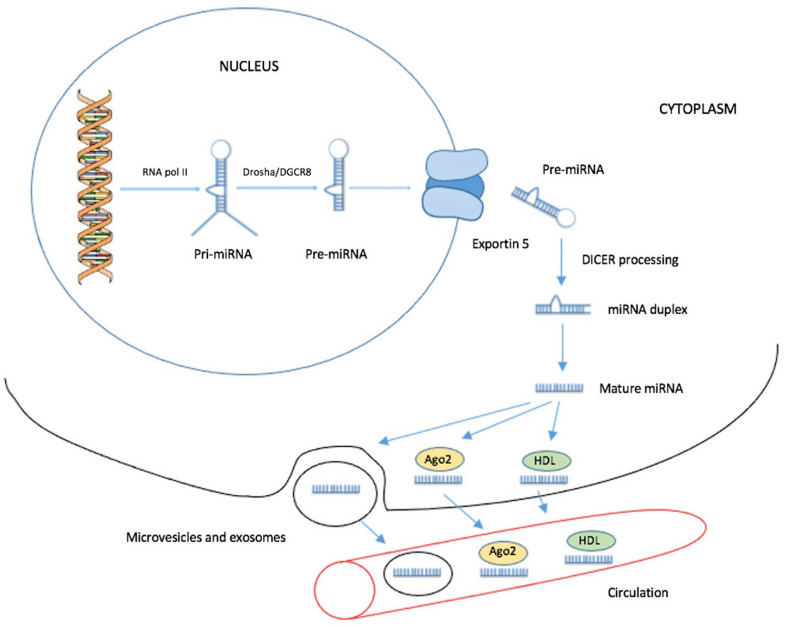
The biogenesis of miRNA is carried out in multiple steps. The miRNA gene is transcribed to generate a primary miRNA (pri-miRNA). After the cleavage of the pri-miRNA to a precursor miRNA (pre-miRNA), the pre-miRNA is exported from the nucleus to the cytoplasm. Last, the pre-miRNA is cleaved to a miRNA duplex, a strand of which is loaded on RISC. There are three mechanisms of miRNA releasing into the circulation: (1) secretion within exosomes, (2) secretion within a complex with the Ago2 protein, and (3) secretion within a complex with HDL.

**Figure 2 biomedicines-09-01468-f002:**
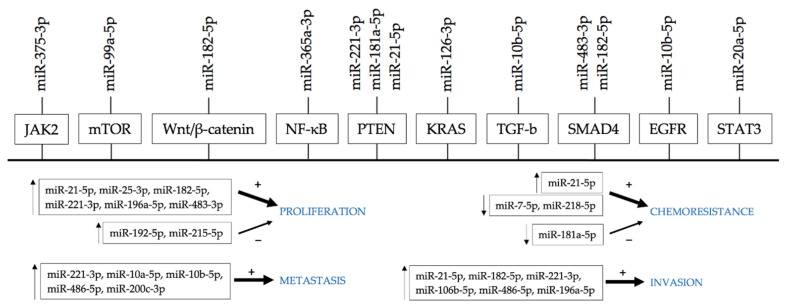
Many miRNAs physiologically regulate the activity of oncogenes and tumor suppressor genes at balance. Up- (↑) or downregulation (↓) of particular miRNAs affects the activity of crucial signaling pathways and cancer hallmarks, such as sustained cell proliferation, invasiveness, and metastasis. Deregulated miRNAs are also involved in chemoresistance.

**Table 1 biomedicines-09-01468-t001:** Circulating miRNAs with a diagnostic significance in early-stage PDAC.

miRNAs with Promising Diagnostic Utility (as Standalone or in Panels)	Source	Control	Number of PDAC Patients: All Stages (Stages I and II) vs. Controls	Regulation in PDAC Samples	Reference
miR-21-5p, miR-30c-5p, miR-10b-5p	Plasma	NC	225 (N/A) vs. 225	Upregulation	[55]
miR-10b-5p, miR-21-5p, miR-30c-5p, miR-181a-5p, let-7a-5p	Plasma/exosomes	CP/NC	29 (27) vs. 17	Upregulation (downregulation: let-7a-5p)	[58]
miR-22-3p, miR-642b-3p, miR-885-5p	Plasma	NC	35 (33) vs. 15	Upregulation	[57]
miR-196a-5p, miR-196b-5p	Serum	pNET/CP/PanIN1/NC	19 (9) vs. 35	Upregulation	[59]
miR-20a-5p, miR-21-5p, miR-24-3p, miR-25-3p, miR-99a-5p, miR-185-5p, miR-191-5p	Serum	CP/NC	197 (74) vs. 240	Upregulation	[60]
miR-642b-3p, miR-885-5p, miR-22-3p	Plasma	NC	11 (11) vs. 22	Upregulation	[61]
miR-486-5p, miR-126-3p, miR-938	Plasma	NC	156 (113) vs. 65	Upregulation	[62]
miR-126-3p, miR-26b-3p, miR-938, miR-19b-3p	Plasma	pNET	156 (113) vs. 27	Upregulation	[62]
miR-486-5p, miR-126-3p, miR-938, miR-663b, miR-19b-3p	Plasma	CP	156 (113) vs. 57	Upregulation (downregulation: miR-663b)	[62]
miR-215-5p, miR-122-5p, miR-192-5p, miR-30b-5p, miR-320b	Serum	CP/NC	50 (15) vs. 75	Upregulation (downregulation: miR-30b-5p and miR-320b)	[63]
miR-21-5p	Plasma	NC	32 (N/A) vs. 42	Upregulation	[76]
miR-483-3p	Plasma	IPMN/NC	32 (N/A) vs. 42	Upregulation	[76]
miR-25-3p	Serum	NC	80 (42) vs. 91	Upregulation	[78]
miR-182-5p	Plasma	CP/NC	109 (38) vs. 88	Upregulation	[83]
miR-221-3p	Plasma	NC	42 PDAC + 5 other pancreatic cancers (16 at stage I-IIA) vs. 30	Upregulation	[48]
miR-10b-5p	Plasma	CP/NC	17 (N/A) vs. 25	Upregulation	[89]

CP: chronic pancreatitis; NC: normal controls; IPMN: intraductal papillary mucinous neoplasm; PanIN1: pancreatic intraepithelial neoplasia, grade 1; pNET: pancreatic neuroendocrine tumor; N/A: not available.

**Table 2 biomedicines-09-01468-t002:** Circulating miRNAs with higher levels and prognostic significance in early-stage PDAC.

miRNA	Tumor Stage	Source	Prognosis	Reference
miR-365a-3p	Resectable	Plasma	Good	[19]
miR-99a-5p	Resectable	Plasma	Good	[19]
miR-200c-3p	Resectable	Plasma	Poor	[19]
miR-21-5p	All	Plasma/serum	Poor	[76,103]
	Resectable	Plasma	Poor	[104]
miR-182-5p	Resectable	Plasma	Poor	[83]
miR-221-3p	All	Plasma	Poor	[48]
miR-375-3p	Resectable	Plasma	Poor	[104]
miR-196a-5p	All	Serum	Poor	[105]

## Data Availability

Not applicable.

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
