# Peer review of "The Role of Circulating MicroRNAs in Patients with Early-Stage Pancreatic Adenocarcinoma"

_biomedicines, 2021, doi:10.3390/biomedicines9101468_

Round 1

Reviewer 1 Report

In the current manuscript the authors have summarized the studies  which sued circulating miRNAs as molecular biomarkers for early diagnosis, prognosis, and chemoresistance in PDAC. The review is well written and cover majority of work done in the field.

However few things which will help to improve the qaulity of the review 

  1.  The language needs to be refined further. Many sentences are confusing .
  2. The introduction is lengthly and redundant. Authors can shorten it bit
  3. Figure 2 doesn't add much to meaning and can be removed 
  4. Reference at some places are very confusing. ex: For Xue et al, line 210, authors have mentioned 64-67. Please add references appropriately, Line 349 authors have used " 3 studies " please add appropriate references. Line 434, authors have added two reference for Gisel et al  (114 and 115). Please edit the references accordingly.

Author Response

REVIEWER’S Comments and Corresponding Responses

Reviewer #1 (Comments to the Author):

 In the current manuscript, the authors have summarized the studies which sued circulating miRNAs as molecular biomarkers for early diagnosis, prognosis, and chemoresistance in PDAC. The review is well written and covers the majority of work done in the field. However few things which will help to improve the quality of the review: 1.      The language needs to be refined further. Many sentences are confusing. We thank the Reviewer for this suggestion. We carefully revised and corrected language and syntax errors. 2.      The introduction is lengthy and redundant. Authors can shorten it a bit.

We conformed with the Reviewer’s comment. The revised introduction is shortened.

 3.      Figure 2 doesn't add much to meaning and can be removed. We complied with the Reviewer’s remark. Previous Figure 2 has been removed. 4.      Reference at some places are very confusing. ex: For Xue et al, line 210, authors have mentioned 64-67. Please add references appropriately, Line 349 authors have used " 3 studies " please add appropriate references. Line 434, authors have added two reference for Gisel et al (114 and 115). Please edit the references accordingly.

Prompted by the Reviewer’s comment, we added the appropriate references, as presented below:

Page 5 (lines 210-211): These results demonstrate that this panel could have non-invasive diagnostic biomarker utility in early-stage PDAC [60].

Page 8 (lines 339-340): Expression of miR-21-5p was measured by either qPCR [60,65,93-99] or in situ hybridization [100-102].

Page 10 (lines 423-424): Gisel et al. observed dysregulation of miR-181a-5p and miR-218-5p in chemoresistant PDAC cell lines with P-glycoprotein overexpression [113].

We also added the respective references:

  1. Liu, R.; Chen, X.; Du, Y.; Yao, W.; Shen, L.; Wang, C.; Hu, Z.; Zhuang, R.; Ning, G.; Zhang, C., et al. Serum microRNA expression profile as a biomarker in the diagnosis and prognosis of pancreatic cancer. Clin Chem 2012, 58, 610-618, doi:10.1373/clinchem.2011.172767.
  2. Nagao, Y.; Hisaoka, M.; Matsuyama, A.; Kanemitsu, S.; Hamada, T.; Fukuyama, T.; Nakano, R.; Uchiyama, A.; Kawamoto, M.; Yamaguchi, K., et al. Association of microRNA-21 expression with its targets, PDCD4 and TIMP3, in pancreatic ductal adenocarcinoma. Mod Pathol 2012, 25, 112-121, doi:10.1038/modpathol.2011.142.
  3. Dhayat, S.A.; Abdeen, B.; Kohler, G.; Senninger, N.; Haier, J.; Mardin, W.A. MicroRNA-100 and microRNA-21 as markers of survival and chemotherapy response in pancreatic ductal adenocarcinoma UICC stage II. Clin Epigenetics 2015, 7, 132, doi:10.1186/s13148-015-0166-1.
  4. Giovannetti, E.; Funel, N.; Peters, G.J.; Del Chiaro, M.; Erozenci, L.A.; Vasile, E.; Leon, L.G.; Pollina, L.E.; Groen, A.; Falcone, A., et al. MicroRNA-21 in pancreatic cancer: correlation with clinical outcome and pharmacologic aspects underlying its role in the modulation of gemcitabine activity. Cancer Res 2010, 70, 4528-4538, doi:10.1158/0008-5472.CAN-09-4467.
  5. Hwang, J.H.; Voortman, J.; Giovannetti, E.; Steinberg, S.M.; Leon, L.G.; Kim, Y.T.; Funel, N.; Park, J.K.; Kim, M.A.; Kang, G.H., et al. Identification of microRNA-21 as a biomarker for chemoresistance and clinical outcome following adjuvant therapy in resectable pancreatic cancer. PLoS One 2010, 5, e10630, doi:10.1371/journal.pone.0010630.
  6. Jamieson, N.B.; Morran, D.C.; Morton, J.P.; Ali, A.; Dickson, E.J.; Carter, C.R.; Sansom, O.J.; Evans, T.R.; McKay, C.J.; Oien, K.A. MicroRNA molecular profiles associated with diagnosis, clinicopathologic criteria, and overall survival in patients with resectable pancreatic ductal adenocarcinoma. Clin Cancer Res 2012, 18, 534-545, doi:10.1158/1078-0432.CCR-11-0679.
  7. Ma, M.Z.; Kong, X.; Weng, M.Z.; Cheng, K.; Gong, W.; Quan, Z.W.; Peng, C.H. Candidate microRNA biomarkers of pancreatic ductal adenocarcinoma: meta-analysis, experimental validation and clinical significance. J Exp Clin Cancer Res 2013, 32, 71, doi:10.1186/1756-9966-32-71.
  8. Papaconstantinou, I.G.; Manta, A.; Gazouli, M.; Lyberopoulou, A.; Lykoudis, P.M.; Polymeneas, G.; Voros, D. Expression of microRNAs in patients with pancreatic cancer and its prognostic significance. Pancreas 2013, 42, 67-71, doi:10.1097/MPA.0b013e3182592ba7.
  9. Wang, P.; Zhuang, L.; Zhang, J.; Fan, J.; Luo, J.; Chen, H.; Wang, K.; Liu, L.; Chen, Z.; Meng, Z. The serum miR-21 level serves as a predictor for the chemosensitivity of advanced pancreatic cancer, and miR-21 expression confers chemoresistance by targeting FasL. Mol Oncol 2013, 7, 334-345, doi:10.1016/j.molonc.2012.10.011.
  10. Dillhoff, M.; Liu, J.; Frankel, W.; Croce, C.; Bloomston, M. MicroRNA-21 is overexpressed in pancreatic cancer and a potential predictor of survival. J Gastrointest Surg 2008, 12, 2171-2176, doi:10.1007/s11605-008-0584-x.
  11. Kadera, B.E.; Li, L.; Toste, P.A.; Wu, N.; Adams, C.; Dawson, D.W.; Donahue, T.R. MicroRNA-21 in pancreatic ductal adenocarcinoma tumor-associated fibroblasts promotes metastasis. PLoS One 2013, 8, e71978, doi:10.1371/journal.pone.0071978.
  12. Khan, K.; Cunningham, D.; Peckitt, C.; Barton, S.; Tait, D.; Hawkins, M.; Watkins, D.; Starling, N.; Rao, S.; Begum, R., et al. miR-21 expression and clinical outcome in locally advanced pancreatic cancer: exploratory analysis of the pancreatic cancer Erbitux, radiotherapy and UFT (PERU) trial. Oncotarget 2016, 7, 12672-12681, doi:10.18632/oncotarget.7208.
  13. Gisel, A.; Valvano, M.; El Idrissi, I.G.; Nardulli, P.; Azzariti, A.; Carrieri, A.; Contino, M.; Colabufo, N.A. miRNAs for the detection of multidrug resistance: overview and perspectives. Molecules 2014, 19, 5611-5623, doi:10.3390/molecules19055611.

Reviewer 2 Report

This is a well-written review on an interesting and important topic, specifically, the role of circulating micro RNA's in patients with curable (non-advanced) pancreatic cancer. It is well-researched and presented.

Due to the emerging role of Artificial Intelligence in healthcare, it would be worthwhile if the authors could comment on the role of AI in the early detection of pancreatic cancer.

Here is a review article on the topic:

Kenner, B., Chari, S. T., Kelsen, D., Klimstra, D. S., Pandol, S. J., Rosenthal, M., Rustgi, A. K., Taylor, J. A., Yala, A., Abul-Husn, N., Andersen, D. K., Bernstein, D., Brunak, S., Canto, M. I., Eldar, Y. C., Fishman, E. K., Fleshman, J., Go, V., Holt, J. M., Field, B., … Wolpin, B. (2021). Artificial Intelligence and Early Detection of Pancreatic Cancer: 2020 Summative Review. Pancreas50(3), 251–279. https://doi.org/10.1097/MPA.0000000000001762

Author Response

REVIEWER’S Comment and Corresponding Response

Reviewer #2 (Comments to the Author):

 This is a well-written review on an interesting and important topic, specifically, the role of circulating micro RNA's in patients with curable (non-advanced) pancreatic cancer. It is well-researched and presented. Due to the emerging role of Artificial Intelligence in healthcare, it would be worthwhile if the authors could comment on the role of AI in the early detection of pancreatic cancer.. We sincerely thank the Reviewer for the positive appraisal of our review article. In response to his remark, we commented on the role of Artificial Intelligence in the Introduction section, as presented here below: Page 2 (lines 57-60): Currently, attempts have been made towards the incorporation of artificial intelligence (AI) tools to further support the PDAC early detection efforts; however, this field is still in its infancy and requires multidisciplinary approaches to evolve [6]. We also added a relevant reference:

  1. Kenner, B.; Chari, S.T.; Kelsen, D.; Klimstra, D.S.; Pandol, S.J.; Rosenthal, M.; Rustgi, A.K.; Taylor, J.A.; Yala, A.; Abul-Husn, N., et al. Artificial Intelligence and Early Detection of Pancreatic Cancer: 2020 Summative Review. Pancreas 2021, 50, 251-279, doi:10.1097/MPA.0000000000001762.

Round 2

Reviewer 2 Report

Thank you for addressing my concerns.